

# Exosomes and ferroptosis: roles in tumour regulation and new cancer therapies

Yixin Shi[1], Bingrun Qiu[1], Linyang Huang[1], Jie Lin[1], Yiling Li[1], Yiting Ze[1], Chenglong Huang[2] and Yang Yao[1]

[1] State Key Laboratory of Oral Diseases, National Clinical Research Center for Oral Diseases, Department of Oral Implantology, West China Hospital of Stomatology, Sichuan University, Chengdu, China
[2] Department of Oral and Maxillofacial Surgery, The Affiliated Stomatology Hospital of Southwest Medical University, Luzhou, China

## ABSTRACT

Research on the biological role of exosomes is rapidly developing, and recent evidence suggests that exosomal effects involve ferroptosis. Exosomes derived from different tissues inhibit ferroptosis, which increases tumour cell chemoresistance. Therefore, exosome-mediated regulation of ferroptosis may be leveraged to design anticancer drugs. This review discusses three pathways of exosome-mediated inhibition of ferroptosis: (1) the Fenton reaction; (2) the ferroptosis defence system, including the Xc-GSH-GPX4 axis and the FSP1/CoQ$_{10}$/NAD(P)H axis; and (3) lipid peroxidation. We also summarize three recent approaches for combining exosomes and ferroptosis in oncology therapy: (1) promoting exosome-inhibited ferroptosis to enhance chemotherapy; (2) encapsulating exosomes with ferroptosis inducers to inhibit cancers; and (3) developing therapies that combine exosomal inhibitors and ferroptosis inducers. This review will contribute toward establishing effective cancer therapies.

Corresponding author
Yang Yao, yaoyang9999@126.com

## INTRODUCTION

Exosomes, which are extracellular vesicles secreted by most cells and are present in many body fluids (*Pegtel & Gould, 2019*), have complex biological roles in cancers and promote cancer progression (*Elewaily & Elsergany, 2021*; *Kalluri & LeBleu, 2020*). For example, exosomes maintain proliferative signalling (*Qu et al., 2009*), activate invasion and metastasis (*Zarin et al., 2021*), induce angiogenesis (*Li et al., 2021a*), and suppress cell death (*Zeng et al., 2020*). Exosomes also enhance tumour cell resistance to radiotherapy and chemotherapy, thereby reducing cancer treatment efficacy (*Hu et al., 2019*). The exosomal regulation of cancer involves multiple mechanisms, which include the ferroptosis regulation (*Brown et al., 2019*). Ferroptosis is a newly identified iron-dependent regulated cell death (RCD), which is caused by massive lipid peroxidation–mediated membrane damage (*Chen et al., 2021d*). Inhibition of ferroptosis promotes cancer progression (*Xu et al., 2020*; *Zhang et al., 2021a*). Ferroptosis regulation strategies have been applied in radiotherapy (*Zhang et al., 2021d*) and chemotherapy (*Niu et al., 2021*) approaches for cancers.

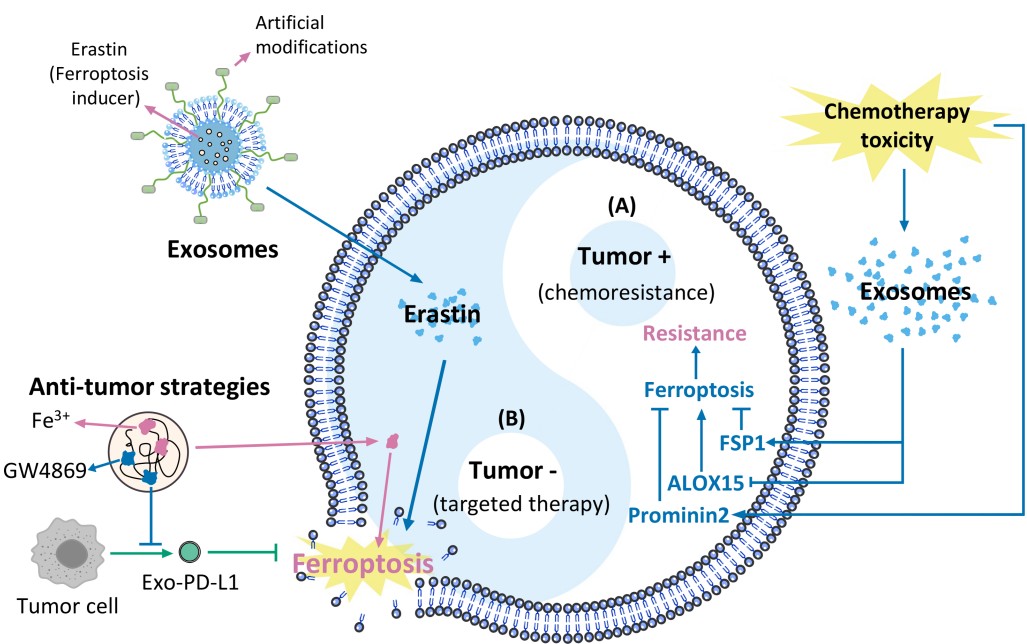

**Figure 1  Exosomes and ferroptosis in tumours.** (A) Mechanisms causing chemoresistance, (B) artificially modified agents targeting cancer therapies.

This review discusses the effects of exosomes on tumour biological behaviours. We focus on three pathways that mediate exosomal actions on ferroptosis, including the Fenton reaction, the ferroptosis defence system (the Xc-GSH-GPX4 axis, and the FSP1/CoQ$_{10}$/NAD(P)H axis), and lipid peroxidation. We propose strategies for applying these pathways to develop cancer therapies. This review summarizes innovative new strategies for reducing tumour chemoresistance and developing more effective exosome-based cancer treatment strategies that target ferroptosis (Fig. 1).

## Why this review is needed and who it is intended for

Exosomes have gained interest recently because of their biological roles in cancer. Recent work has identified a complex relationship between exosomes and ferroptosis. However, no review summarizes in detail the regulatory pathways of cell-derived exosomes on ferroptosis and their potential for cancer therapy. This review discusses the effects of exosomes on tumour cell biology and summarizes three pathways of exosome-ferroptosis regulation. We also propose therapeutic strategies based on the exosome-ferroptosis effect. Our review will appeal to researchers interested in ferroptosis and exosomes, providing them with innovative ideas and insights for future experiments, as well as an overview of research on the combination of exosomes and ferroptosis in tumour therapy.

## SURVEY METHODOLOGY

We conducted a systematic search of the literature to identify relevant articles for this review using PubMed, Web of Science, and Google Scholar, with the last search conducted on

March 5, 2022. The search was performed in full-text journals, focusing on the regulatory pathways of exosomes on ferroptosis and their role in cancers. The keywords used and their synonyms and variants could be classified into categories and any combination of words from different categories was used for the search. The categories we used are as follows:

1. About exosomes: exosomes; exosomal; extracellular vesicles (EVs); exosomal biosynthesis; secretion; uptake; endocytosis
2. About ferroptosis: ferroptosis; anti-ferroptosis; ferroptosis mechanisms; lipid peroxidation; Fenton reaction; arachidonic acid lipoxygenases (ALOXs); glutathione (GSH); glutathione peroxidase 4 (GPX4); ferroptosis suppressor protein 1 (FSP1); GTP cyclohydrolase 1 (GCH1); BH4; solute carrier family 7 member 11 (SLC7A11); solute carrier family 3 member 2 (SLC3A2); ferroptosis inducers; ferritinophagy; ferroptosis defence; System Xc-
3. About tumour: tumour; cancer; anticancer; antitumour; tumourigenesis; invasion; migration; cell proliferation; angiogenesis; metastasis; inflammatory; cell death; apoptosis; chemoresistance; radioresistance; antimicrobial death; immune escape; immunosuppression.

The words were merged *via* the Boolean operators 'AND' and 'OR'. The initial search screened approximately 600 relevant articles written in English that could be useful for this review.

## The biological role of exosomes in tumours

Exosomal cargoes include RNA, DNA, proteins, carbohydrates, and lipids. The RNA species include mRNAs, long non-coding RNAs (lncRNAs), and microRNAs (miRNAs) (*O'Brien et al., 2020*). The biological roles of exosomes have become a topic of interest (*Kalluri & LeBleu, 2020*), especially the role of exosomes in tumour development and cancer progression (*Elewaily & Elsergany, 2021*; *Pi et al., 2021*). The regulation of biological tumour phenotypes by exosomes has mainly focused on lncRNAs and microRNAs, followed by proteins and lipids (Table 1).

Tumour-derived exosomes promote tumour formation in non-tumour cells (*Abd Elmageed et al., 2014*; *Melo et al., 2014*) and the proliferation of tumour cells (*Qi, Zhang & Wang, 2021*; *Sun et al., 2019*), both of which promote tumour progression. Tumour-derived exosomes contain miRNAs (*Du et al., 2020*; *Zhou et al., 2019b*), lncRNAs (*Lang et al., 2017a*; *Lang et al., 2017b*), and proteins (*Yamashita et al., 2019*) that promote angiogenesis and increase tumour malignancy after being taken up by vascular endothelial cells. Exosomes secreted by tumour cells or tumour stromal cells regulate the metabolism of the pre-metastatic tumour microenvironment (*Fong et al., 2015*), disrupt the tight junctions of vascular endothelial cells and the vascular endothelial barrier (*Li et al., 2018a*; *Zhou et al., 2014*), and promote tumour invasion and metastasis. Tumour-derived exosomes also alter the distal environment, allowing cancer cells to metastasize at distal sites (*Hoshino et al., 2015*; *Zhang et al., 2017*). Exosomes inhibit apoptosis through miRNA transfer and regulation of apoptosis-associated proteins (*Huang et al., 2019*; *Zeng et al., 2020*). The regulation of inflammation by exosomes in tumour cells has both positive and negative consequences. Some tumour-derived exosomes promote the expression

**Table 1 Regulation of cancer by exosomal cargoes.**

| Type | Cargoes | Cancer | Regulated biological phenotype | Mechanism | Ref |
|---|---|---|---|---|---|
| lncRNA | lncRNA UCA1 | pancreatic cancer | promotes chemoresistance | SOCS3/EZH2 axis | Chi, Xin & Liu (2021) |
| | | vulvar squamous cell carcinoma | promotes chemoresistance | miR-103a/WEE1 axis | Gao et al. (2021a) |
| | | cervical cancer | promotes proliferation, invasion, and migration and inhibits apoptosis | miR-122-5p/SOX2 axis | Gao et al. (2021d) |
| | lncRNA NEAT1 | endometrial cancer | promotes tumourigenesis | miR-26a/b-5p-mediated STAT3/YKL-40 signalling pathway | Fan et al. (2021) |
| | | ovarian cancer | promotes chemoresistance | miR-491-5p/SOX3 axis | Jia, Wei & Zhang (2021) |
| | | prostate cancer | promotes metastasis | miR-205-5p/RUNX2/ SFPQ/ PTBP2 axis | Mo et al. (2021) |
| | lncRNA H19 | hepatocellular carcinoma | promotes proliferation | miR-520a-3p/LIMK1 axis | Wang et al. (2020b) |
| | | non-small cell lung cancer | promotes chemoresistance | miR-615-3p/ATG7 axis | Lei et al. (2018) and Pan & Zhou (2020) |
| | | colorectal cancer | promotes chemoresistance | β-catenin pathway | Ren et al. (2018) |
| | miR-155 | non-small-cell lung cancer | promotes metastasis | targets RASSF4 | Li et al. (2021c) |
| | | breast cancer | promotes invasion | targets PPARγ | Wu et al. (2018) |
| | | hepatocellular carcinoma cell | promotes proliferation | targets PTEN | Sun et al. (2019) |
| | | gastric cancer | promotes angiogenesis | targets FOXO3a | Zhou et al. (2019b) |
| | | | promotes angiogenesis | C-MYB/VEGF axis | Deng et al. (2020) |
| | | pancreatic cancer | promotes chemoresistance | targets DCK | Patel et al. (2017) |
| | | myeloma | promotes proliferation, chemoresistance, and inhibits apoptosis | Hedgehog signalling pathway | Gao et al. (2021b) |
| | miR-155-5p | ovarian cancer | induces immune escape | miR-155-5p/PD-L1 pathway | Li et al. (2022b) |
| | | renal cell carcinoma | promotes proliferation and metastasis | HuR-dependent IGF1R/AKT/PI3K pathway | Gu et al. (2021) |
| | | colon cancer | induces immune escape | ZC3H12B/IL-6 axis | Ma et al. (2021) |
| | | gastric cancer | promotes proliferation and migration | targets TP53INP1 | Shi et al. (2020a) |
| | miR-21 | gastric cancer | promotes chemoresistance and inhibits apoptosis | targets PTEN, PI3K/AKT signalling pathway | Zheng et al. (2017) |
| | | esophageal squamous cell carcinoma | promotes chemoresistance | STAT3 signalling | Zhao et al. (2021) |
| | | esophageal cancer | promotes invasion and migration | targets PDCD4, JNK signalling pathway | Liao et al. (2016) |
| | | non-small-cell lung cancer | promotes chemoresistance | targets PTEN | Dong et al. (2019) |
| | | hepatocellular carcinoma | promotes proliferation and metastasis | TETs/PTENp1/PTENv pathway | Chi, Xin & Liu (2021), Chi, Xin & Liu (2021), Cao et al. (2019) and Tian et al. (2019) |

**Table 1** (*continued*)

| Type | Cargoes | Cancer | Regulated biological phenotype | Mechanism | Ref |
|---|---|---|---|---|---|
| | | osteosarcoma | promotes proliferation and invasion | targets PIK3R1, PI3K/Akt/mTOR pathway | *Qi, Zhang & Wang (2021)* |
| | miR-21-5p | ovarian cancer | promotes invasion and migration | targets CDK6 | *Cao et al. (2021)* |
| | | gastric cancer | promotes metastasis | targets SMAD7, TGF-β/Smad pathway | *Li et al. (2018b)* |
| | | gastric cancer | promotes angiogenesis | targets PTEN, AKT pathway | *Du et al. (2020)* |
| | | hepatocellular carcinoma | promotes migration and chemoresistance | VHL/HIF axis | *Liu et al. (2019)* |
| | miR-23a | nasopharyngeal carcinoma | promotes angiogenesis | targets TSGA10 | *Bao et al. (2018)* |
| | | lung cancer | Promotes angiogenesis and migration | targets ZO-1, PHD1 and 2/HIF-1α axis | *Hsu et al. (2017)* |
| | | lung cancer | promotes proliferation and invasion | RUNX3/PI3K/AKT signalling pathway axis | *Li, Chen & Yi (2021e)* |
| | | non-small cell lung cancer | promotes proliferation, migration, and invasion | PTEN/PI3K/AKT pathway | *Yang et al. (2020a)* |
| | miR-210 | pancreatic cancer | promotes chemoresistance | activates mTOR signalling | *Yang et al. (2020b)* |
| | | hepatocellular carcinoma | promotes angiogenesis | targets SMAD4 and STAT6 | *Lin et al. (2018)* |
| | | colorectal cancer | promotes proliferation and inhibits apoptosis | targets CELF2 | *Ge et al. (2021)* |
| | miR-210-3p | lung cancer | promotes migration and invasion | targets FGFRL1 | *Wang et al. (2020d)* |
| | | oral squamous cell carcinoma | promotes angiogenesis | EFNA3/PI3K/AKT pathway | *Wang et al. (2020c)* |
| | | breast cancer | promotes proliferation, invasion and chemoresistance | targets CCNG2 | *Li et al. (2017)* |
| | miR-1246 | glioma | promotes migration and invasion | targets FRK | *Qian et al. (2021)* |
| | | oral squamous cell carcinoma | promotes invasion | targets DENND2D | *Sakha et al. (2016)* |
| | | prostate cancer | promotes chemoresistance | targets GREM2, TGF-β signalling pathway | *Shan et al. (2020)* |
| | miR-423-5p | breast cancer | promotes chemoresistance | targets P-glycoprotein | *Wang et al. (2019a)* |
| | | gastric cancer | promotes metastasis | targets SUFU | *Yang et al. (2018a)* |

| Type | Cargoes | Cancer | Regulated biological phenotype | Mechanism | Ref |
|---|---|---|---|---|---|
| miRNA | | pancreatic ductal adenocarcinoma | promotes proliferation and invasion | PPP2R2A/AKT/p27 axis | *Li et al. (2018c)* |
| | | breast cancer | promotes proliferation, migration and invasion | targets PTEN, Akt pathway | *Chen et al. (2021b)* |
| | | colon cancer | promotes proliferation, migration, and metastasis | targets MIA3 | *Du et al. (2021b)* |
| | miR-222 | melanoma | promotes invasion | PI3K/AKT pathway | *Felicetti et al. (2016)* |
| | | colorectal cancer | promotes metastasis | targets SPINT1, SPINT1/HGF axis | *Tian et al. (2021)* |
| | miR-221 | colorectal cancer | promotes metastasis | targets SPINT1, SPINT1/HGF axis | *Tian et al. (2021)* |
| | | glioma | promotes chemoresistance | targets DNM3 | *Yang et al. (2017)* |
| | | oral squamous cell carcinoma | promotes migration and angiogenesis | targets PIK3R1 | *He et al. (2021)* |
| | | osteosarcoma | promotes the growth and metastasis | SOCS3/JAK2/STAT3 axis | *Liu et al. (2021c)* |
| | miR-221-3p | cervical squamous cell carcinoma | promotes angiogenesis | targets THBS2 | *Wu et al. (2019)* |
| | | | promotes lymphangiogenesis and metastasis | targets VASH1 | *Zhou et al. (2019a)* |
| | | epithelial ovarian cancers | promotes proliferation | targets CDKN1B | *Li & Tang (2020)* |
| | | lung adenocarcinoma | promotes metastasis | Hippo pathway | *Chen et al. (2021a)* |
| | | esophageal cancer | promotes migration and invasion, inhibits apoptosis | targets PTEN | *Zeng et al. (2020)* |
| | miR-19b-3p | | promotes proliferation, migration, invasion, and inhibits apoptosis | targets SOCS1 | *Deng et al. (2021)* |
| | | clear cell renal cell carcinoma | promotes metastasis | targets PTEN | *Wang et al. (2019b)* |
| | | colon cancer | promotes chemoresistance | CDX2/HEPH axis | *Zhang et al. (2021b)* |
| | miR-24-3p | oral squamous cell carcinoma | promotes proliferation | targets PER1 | *He et al. (2020)* |
| | | nasopharyngeal carcinoma | induces immune escape | targets FGF11 | *Ye et al. (2016)* |
| lipid | FAs | breast cancer | induces immune escape | PPARα signalling | *Yin et al. (2020)* |
| | | cervical carcinoma | induces immune escape | PPARα signalling | *Yin et al. (2020)* |
| | | melanoma | induces immune escape | PPARα signalling | *Yin et al. (2020)* |
| | | non-small cell lung cancer | promotes chemoresistance | PI3K/AKT and MAPK pathways | *Wu et al. (2021a)* |

**Table 1** (*continued*)

| Type | Cargoes | Cancer | Regulated biological phenotype | Mechanism | Ref |
|---|---|---|---|---|---|
| protein | EGFR | gastric cancer | promotes liver metastasis | miR-26a and b/HGF pathway | *Zhang et al. (2017)* |
| | | oral squamous cell carcinoma | promotes invasion | – | *Fujiwara et al. (2018)* |
| | | non-small cell lung cancer | induces immune escape | PD-1/PD-L1 pathway | *Kim et al. (2019)* |
| | PD-L1 | melanoma | induces immunosuppression | PD-1/PD-L1 pathway | *Chen et al. (2018)* |
| | | breast cancer | induces immunosuppression | PD-1/PD-L1 pathway | *Yang et al. (2018b)* |
| | | head and neck squamous cell carcinomas | induces immunosuppression | PD-1/PD-L1 pathway | *Theodoraki et al. (2018)* |
| | EphA2 | lung cancer | promotes angiogenesis | MAPK signalling | *Yamashita et al. (2019)* |
| | | pancreatic cancer | promotes chemoresistance | – | *Fan et al. (2018)* |
| | | breast cancer | promotes metastasis | EphA2-Ephrin A1 reverse signalling | *Gao et al. (2021c)* |

**Notes.**

UCA1, urothelial carcinoma-associated 1; SOCS3, suppressor of cytokine signalling 3; EZH2, enhancer of zeste homolog 2; WEE1, WEE1 G2 checkpoint kinase; SOX2, sex determining region Y box 2; NEAT1, nuclear enriched abundant transcript 1; STAT3, signal transducer and activator of transcription 3; YKL-40, chitinase 3-like protein 1; SOX3, sex determining region Y box 3; RUNX, runt-related transcription factor 2; SFPQ, splicing factor proline and glutamine-rich; PTBP2, polypyrimidine-tract-binding protein 2; LIMK1, LIM domain kinase 1; ATG7, autophagy-associated gene 7; RASSF4, ras association domain family member 4; PPAR $\gamma$, peroxisome proliferator-activated receptor gamma; PTEN, phosphatase and tensin homolog; FOXO3a, forkhead Box O3a; VEGF, vascular endothelial growth factor; DCK, deschloroketamine;; PD-L1, programmed cell death ligand 1; IGF1R, Insulin-like growth factor 1 receptor; AKT, protein kinase B; PI3K, phosphoinositide 3-kinase; ZC3H12B, zinc finger CCCH-type-containing 12B; IL-6, interleukin 6; TP53INP1, tumour protein 53-inducesd nuclear protein 1; PTEN, phosphatase and tensin homolog; PDCD4, programmed cell death 4; JNK, c-Jun N-terminal kinase; TETs, Tet methylcytosine dioxygenases; PTENp1, phosphatase and tensin homolog pseudogene 1; PIK3R1, phosphoinositide-3-kinase regulatory subunit 1; mTOR, rapamycin; CDK6, cyclin-dependent kinase 6; SMAD7, drosophila mothers against the decapentaplegic 7; TGF-β, transforming growth factor-β; VHL, von Hippel-Lindau; HIF, hypoxia-inducible factor; TSGA10, testis-specific gene antigen 10; ZO-1, zonula occludens-1; PHD, prolyl hydroxylases; RUNX3, runt-related transcription factor 3; SMAD4, drosophila mothers against the decapentaplegic 4; STAT6, signal transducer and activator of transcription 6; CELF2, CUGBP Elav-like family member 2; FGFRL1, fibroblast growth factor receptor-like 1; EFNA3, ephrin A3; CCNG2, cyclin G2; FRK, fructokinase; DENND2D, DENN/MADD Domain Containing 2D; GREM2, gremlin-2; SUFU, suppressors-of-fused homolog; PPP2R2A, phosphatase protein phosphatase 2 regulatory subunit βα; MIA3, melanoma inhibitsory activity member 3; SPINT1, serine peptidase inhibitsor, Kunitz type -1; HGF, hepatocyte growth factor; DNM3, dynamin 3; JAK2, janus kinase 2; THBS2, thrombospondin 2; VASH1, vasohibin-1; CDKN1B, cyclin-dependent kinase inhibitsor 1B; SOCS1, suppressor of cytokine signalling 1; CDX2, caudal type homeobox 2; HEPH, hephaestin; PER1, period circadian regulator 1; FGF11, fibroblast growth factor 11; FAs, fatty acids; PPARα, proliferator activated receptor α, peroxisome; EGFR, epidermal growth factor receptor; MAPK, mitogen-activated protein kinase; PD-1, programmed death-1; EphA2, ephrin-A receptor 2.

of inflammatory mediators, thereby promoting cellular inflammatory responses and tumour progression (*Chow et al., 2014*; *Wu et al., 2016*). In contrast, other tumour-derived exosomes attenuate tumour inflammation and promote the immune escape of cancer cells (*Othman, Jamal & Abu, 2019*). Among these exosomes, programmed cell death 1 (PD-1) and its ligand (PD-L1) are the most well studied (*Xie et al., 2019*). By binding the PD-1 receptor expressed on activated T cells, PD-L1 inhibits the activation and proliferation of T cells, thereby protecting tumour cells from being killed by T cells and leading to immune escape (*Chen et al., 2018*; *Lawler et al., 2020*). Exosomes secreted by tumour cells and cancer-associated fibroblasts (CAFs) promote tumour cell chemoresistance through the delivery of exosomal cargo (*Hu et al., 2019*; *Yang et al., 2020b*). A recent study proposes that the CAF-derived exosome miR-522 inhibits ferroptosis by inhibiting the activity of the arachidonate 15-lipoxygenase (ALOX15) and reducing lipid reactive oxygen species (ROS) accumulation and lipid peroxidation, thereby promoting chemoresistance (*Zhang et al., 2020*). In addition, exosome-mediated ferroptosis inhibition is a novel mechanism for gastric cancer (GC)-acquired chemoresistance. The mechanism of ferroptosis inhibition will be described in subsequent sections.
## Ferroptosis regulation in tumours

Ferroptosis is a form of RCD characterized by ROS accumulation and lipid peroxidation (*Dixon et al., 2012*). The cessation of lipid peroxide removal triggers ferroptosis (Fig. 2).

One of the primary ferroptosis mechanisms includes enzymatic and nonenzymatic lipid peroxidation. Enzymatic lipid peroxidation is an oxidative reaction that occurs in the presence of ALOXs, whereas nonenzymatic lipid peroxidation is driven by iron and ROS-induced free radicals *via* the Fenton reaction (*Chen et al., 2021d*; *Jiang, Stockwell & Conrad, 2021*; *Tang et al., 2021*). Downregulation of ALOX15 expression and enzymatic lipid production promotes cancer progression (*Tian et al., 2017*), and activation of ALOX15 in cancer cells inhibits cancer growth (*Weigert et al., 2018*). ALOX15 catalyses enzymatic lipid peroxidation, suggesting that ALOX15 may inhibit tumours by promoting ferroptosis. Another enzyme associated with lipid peroxidation, stearoyl-CoA desaturase 1 (SCD1), promotes anti-ferroptosis and tumour growth in gastric cancer cells (*Wang et al., 2020a*).

Iron metabolism, including $Fe^{3+}$ input, Fe reaction, and $Fe^{2+}$ output, is another mechanism involved in ferroptosis. Ferritin is the main site of iron storage in the cell, and consists of ferritin light chain (FTL) and ferritin heavy chain 1 (FTH1) (*Muhoberac & Vidal, 2019*). Iron metabolism may change in tumour cells and promote the initiation and growth of cancer. The lncRNA RP11-89 sponges miR-129-5p and upregulates prominin2 (prom2), thereby promoting iron export and inhibiting ferroptosis to facilitate tumourigenesis (*Luo et al., 2021*).

Anti-ferroptosis mechanisms in tumours promote cancer progression. Ferroptosis defence systems consist of three signalling axes: the Xc–GSH-GPX4 axis (*Dixon et al., 2012*), the FSP1/$CoQ_{10}$/NAD(P)H axis (*Bersuker et al., 2019*; *Doll et al., 2019*), and the GCH1-BH4 axis (*Kraft et al., 2020*; *Stockwell, Jiang & Gu, 2020*). GPX4 uses GSH as its cofactor to transform phospholipid hydroperoxide (puFA-PL-OOH) into nontoxic phospholipid alcohol (puFA-PL-OH), reducing the accumulation of toxic lipid peroxides. Ubiquinol traps lipid peroxyl radicals and suppresses lipid peroxidation. BH4 suppress ferroptosis by aiding the formation of reduced $CoQ_{10}$, and blocking the peroxidation of specific lipids through causing lipid remodelling. The regulatory mechanisms of ferroptosis vary in different tumours. In hepatocellular carcinoma (HCC), GPX4 is upregulated by the Circ-interleukin-4 receptor, and GPX4 upregulation suppresses miR-541-3p–induced ferroptosis and promotes tumourigenesis (*Xu et al., 2020*). In GC, the CD44 variant CD44v interacts with the Xc–system, controls the intracellular level of reduced GSH, and promotes GC growth *via* anti-ferroptosis (*Ishimoto et al., 2011*).

Ferroptosis activation inhibits tumour progression. The loss of the $Xc^-$ system in melanoma reduces intracellular cystine levels, decreasing GSH synthesis and GPX4 activity. This process promotes ferroptosis and eliminates tumour metastasis (*Sato et al., 2020*). Gambogenic acid induces ferroptosis *via* the p53/SLC7A11/GPX4 signalling pathway and inhibits melanoma cell migration and epithelial-to-mesenchymal transition (*Wang et al., 2020e*). Drug-resistant cancer cells depend on GPX4 and are more likely to undergo ferroptosis (*Hangauer et al., 2017*; *Tsoi et al., 2018*). Zinc finger E-Box binding homeobox 1 (ZEB1) has high expression levels in several treatment-resistant cancer cell lines that depend on GPX4 and high sensitivity to ferroptosis caused by GPX4

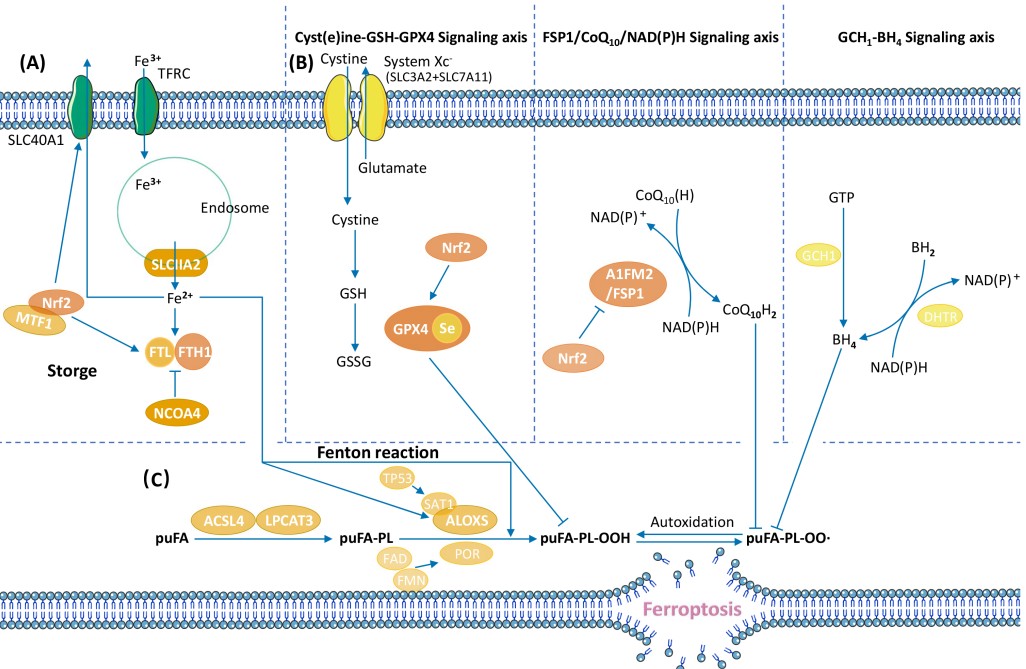

**Figure 2  Overview of ferroptosis pathways.** (A) Iron metabolism, (B) ferroptosis defence systems (Xc-GSH-GPX4 signalling axis, FSP1/CoQ$_{10}$/NAD(P)H signalling axis, and GCH1-BH4 signalling axis), (C) lipid peroxide regulation (*Stockwell et al., 2017*; *Tang et al., 2021*).

inhibition (*Viswanathan et al., 2017*). This suggests that the induction of ferroptosis may be a promising approach for cancer treatment. Induced ferroptosis or ferroptosis inducers combined with chemotherapy or radiotherapy can eliminate and inhibit tumour cells. Erianin induces calcium/calmodulin-dependent ferroptosis and inhibits cancer cell migration, thereby exhibiting anticancer activity (*Chen et al., 2020*). The P62- Kelch like ECH associated protein 1 (KEAP1)- nuclear factor erythroid-2 related factor 2 (Nrf2) pathway has a role in HCC cell ferroptosis, and inhibition of Nrf2 expression upregulates iron and ROS levels and promotes the antitumour effects of ferroptosis inducers (*Sun et al., 2016*). Ferroptosis inducers can be combined with radiotherapy in cancer treatment strategies to suppress radioresistant cancers by inactivating SLC7A11 or GPX4 (*Lang et al., 2019*; *Lei et al., 2020*; *Ye et al., 2020*). In a murine xenograft model and human patient–derived models, ferroptosis inducers enhance the antitumour effects of radiotherapy (*Ye et al., 2020*).

## Mechanism of exosome-mediated ferroptosis

Exosomes transport specialized cargo molecules that regulate the expression of ferroptosis-related genes in receptor cells. Since the regulation of ferroptosis affects tumour development, the mechanism of exosome-mediated ferroptosis must also be investigated (Fig. 3).

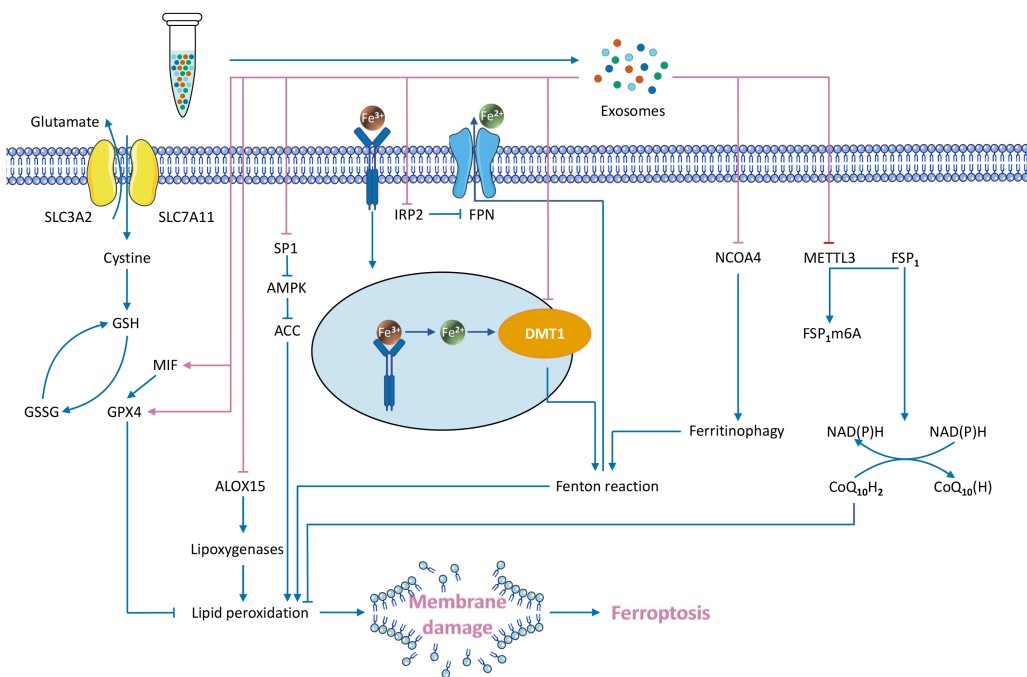

**Figure 3** Exosome-mediated inhibition of ferroptosis and known regulatory mechanisms.

## Exosomes directly inhibit the Fenton reaction

Iron accumulation contributes to ROS production through the Fenton reaction, promoting lipid peroxidation and leading to ferroptosis. Exosomes can reduce the intracellular iron content, which may inhibit the Fenton reaction and subsequent ferroptosis (*Dixon et al., 2012*).

*Inhibition of iron transport reduces the intracellular iron content.* Exosomes downregulate two key genes involved in regulating intracellular iron transport: divalent metal transporter 1 (DMT1) (*Song et al., 2020*) and iron regulatory protein2 (IRP2) (*Yi & Tang, 2021*). $Fe^{3+}$ enters the cell through transferrin receptor (TFRC), is converted to $Fe^{2+}$ in endosomes *via* the metal reductase six-transmembrane epithelial antigen of prostate 3 (Steap3), and is subsequently released from endosomes *via* DMT1 (*Wu et al., 2021b*). This process may facilitate the Fenton reaction and enhance ferroptosis. DMT1 regulates the level of intracellular iron, which is closely associated with ferroptosis (*Hubert & Hentze, 2002*; *Li et al., 2019*; *Núñez & Hidalgo, 2019*; *Yu et al., 2019a*). Reduced DMT1 expression reduces the intracellular iron level (*Du et al., 2016*). DMT1 is the target gene of exosomal miR-23a-3p, and human umbilical cord blood mesenchymal stem cell (HUCB-MSC)–derived exosomes suppress ferroptosis through miR-23a-3p inhibition of DMT1 expression (*Song et al., 2020*). IRP2, another regulator of the intracellular iron content gene, is the target gene of exosomal miR-19b-3p. Exosomes from miR-19b-3p-modified adipose-derived stem cells (ADSCs-19bM-Exos) repress IRP2 expression (*Yi & Tang, 2021*). IRP2 overexpression increases TFRC and decreases ferroportin (FPN, an iron output protein), thereby increasing

intracellular iron content (*Yi & Tang, 2021*). Conversely, reduced IRP2 expression limits intracellular iron transmission in neuronal cells (*Ripa et al., 2017*; *Wang et al., 2014b*). ADSCs-19bM-Exos inhibit ferroptosis. Therefore, the key genes that regulate intracellular iron transport are involved in the mechanisms of exosomal inhibition of ferroptosis.

*Inhibition of ferritinophagy.* Ferritinophagy is a recently discovered form of selective autophagy that regulates intracellular iron metabolism. Excessive activation of ferritinophagy increases the intracellular free iron content and leads to ferroptosis (*Li et al., 2020a*; *Ni et al., 2021*). One of the most prominent ferritinophagy genes is nuclear receptor coactivator 4 (*NCOA4*). The NCOA4 protein binds directly to FTH1 for transport to the autophagosome, which fuses with the lysosomes that degrade ferritin to release iron (*Dowdle et al., 2014*; *Santana-Codina, Gikandi & Mancias, 2021*). Vascular endothelial cell–derived exosomes (EC-Exos) inhibit ferroptosis by inhibiting ferritinophagy (*Yang et al., 2021b*). EC-Exos treatment reversed dexamethasone-induced NCOA4 and autophagy-related protein (including LC3II and beclin-1) upregulation (*Yang et al., 2021b*). NCOA4 overexpression enhances ferroptosis and reduces this protective effect. Downstream of the action of NCOA4, the protein α-synuclein (α-syn) impairs ferritinophagy (*Baksi & Singh, 2017*). α-syn is associated with dysregulation of iron homeostasis and ferroptosis (*Mahoney-Sánchez et al., 2021*). α-syn inhibits ferritin degradation and releases downstream of autophagosome formation, possibly in lysosomes (*Baksi & Singh, 2017*). Lysosomal dysfunction increases exosome-mediated release and delivery of α-synuclein, resulting in prion-like transmission of α-syn (*Alvarez-Erviti et al., 2011*).

### Exosomes activate ferroptosis defence pathways

*Upregulation of GPX4 expression.* GPX4 is a core regulatory node that inhibits ferroptosis. Several studies showed that exosomes upregulate GPX4, thereby inhibiting ferroptosis (*Gan et al., 2021*; *Li et al., 2020b*; *Yang et al., 2021b*). For example, plasma-derived exosomes (RP-Exos) upregulate GPX4 expression and reduce lipid peroxidation in the cell membrane (*Gan et al., 2021*), and EC-Exos upregulates GPX4 in a concentration-dependent manner (*Yang et al., 2021b*). miR-137 in endothelial progenitor cell (EPC)–derived exosomes increased GSH and GPX4, whereas exosomes without miR-137 do not increase the levels of GSH and GPX4 (*Li et al., 2020b*). A recent study has found that nasopharyngeal carcinoma cell-derived exosomes increase GPX4 expression *via* macrophage migration inhibitory factor (MIF) and GPX4 expression is positively correlated with MIF (*Chen et al., 2021c*). In addition, three possible mechanisms underlie exosomal upregulation of GPX4: (1) direct upregulation of GPX4 expression; (2) inhibition of GSH solubilization and increased GPX4 activity (*Li et al., 2020b*; *Ursini & Maiorino, 2020*); and (3) upregulation of Nrf2, which increases GSH levels and GPX4 activity (*Liu et al., 2021d*).

*Upregulation of FSP1 protein expression.* Exosomes inhibit ferroptosis by regulating FSP1, a ferroptosis inhibitor parallel to GPX4 (*Bersuker et al., 2019*; *Doll et al., 2019*). Delivery of cisplatin-resistant non-small-cell lung cancer (NSCLC) exosomal miR-4443 to cisplatin-sensitive NSCLC downregulates the methyltransferase-like 3 (METTL3) gene, thereby upregulating FSP1 mRNA levels and downregulating ferroptosis (*Song et al., 2021b*). In

this system, METTL3 is the target of miR-4443, and this inhibition ultimately upregulates FSP1 and suppresses ferroptosis (*Song et al., 2021b*). Few modulators of FSP1 have been studied. One FSP1 modulator, 8,9-epoxyeicosatrienoic acid (8,9-EET), restores FSP1 expression in pancreatic cancer cells treated with ferroptosis inducers (*Tao et al., 2021*). This study suggests that EETs may inhibit ferroptosis by upregulating FSP1. In contrast, in addition to the well-known FSP1 inhibitor iFSP1 (*Doll et al., 2019*), a compound targeting FSP1 protein, NPD4928, has recently been reported to enhance ferroptosis by inhibiting FSP1 (*Yoshioka et al., 2022*).

### Exosomes modulate ferroptosis by inhibiting other pathways

CAF-derived exosomes inhibit ferroptosis by the miR-522 /ALOX15 axis (*Zhang et al., 2020*). ALOX15 is synthesized *via* the iron-catalysed enzymatic reaction (*Doll & Conrad, 2017*). ALOX15 upregulation causes excess polyethylene hydroperoxides to accumulate beyond the reducing capacity of GPX4, ultimately leading to ferroptosis (*Wenzel et al., 2017*). CAF-derived exosomal miR-522 inhibits ALOX15 in GC cells, reducing the accumulation of lipid ROS and suppressing ferroptosis. Conversely, increased exosomal miR-522 promotes ferroptosis (*Zhang et al., 2020*). This process is regulated at the posttranscriptional level (*Zhang et al., 2020*). EPC-derived exosomes inhibit ferroptosis through the miR-30e-5p/specific protein 1 (SP1)/adenosine monophosphate-activated protein kinase (AMPK) axis (*Xia et al., 2022*). miR-30e-5p targets SP1, and SP1 inhibits activation of the AMPK pathway (*Xia et al., 2022*), which inhibits ferroptosis through phosphorylation of acetyl-CoA carboxylase (ACC) (*Lee et al., 2020*). EPC-derived exosomes upregulate miR-30e-5p, inhibit SP1, and activate the AMPK pathway to inhibit ferroptosis (*Xia et al., 2022*). In addition to miRNAs, exosomal lncRNAs also inhibit ferroptosis. Bone marrow mesenchymal stem cell (BMSC)-derived exosomal lncRNA Mir9-3 host gene (lncRNA Mir9-3hg) inhibits ferroptosis in cardiomyocytes *via* the pumilio RNA binding family member 2 (Pum2)/peroxiredoxin 6 (PRDX6) axis (*Zhang et al., 2022*). LncRNA Mir9-3hg downregulates the expression of Pum2, which binds the PRDX6 promoter to suppress PRDX6 expression (*Zhang et al., 2022*). PRDX6 is a negative regulator of ferroptosis, and specific PRDX6 phospholipase A2 inhibitors enhance ferroptosis (*Lu et al., 2019*). BMSC-derived exosomal lncRNA Mir9-3hg inhibits Pum2 and upregulates PRDX6, thereby suppressing ferroptosis (*Zhang et al., 2022*).

## Molecular mechanisms of exosomes and tumour therapies combining ferroptosis and exosomes

Exosomal regulation of ferroptosis in receptor cells is related to exosome synthesis and uptake mechanisms. There are five key steps in exosomal biosynthesis: (1) endocytosis of the cytoplasmic membrane, (2) early sorting endosomes (ESE), (3) late sorting endosomes (LSE), (4) formation of multivesicular bodies (MVBs) containing future exosomes, and (5) exosome release (*Kalluri & LeBleu, 2020*; *Pan & Johnstone, 1983*). These processes involve a variety of proteins and lipids. For example, the Endosomal sorting complex required for transport (ESCRT) proteins bind in a continuous complex (ESCRT-0, -I, -II, and -III) across the MVB membrane to regulate cargo orientation and the formation of intraluminal vesicles (ILVs) (*Hurley, 2015*). The transmembrane Tetraspanin proteins
induce membrane-bending structures and promote exosome formation (*Andreu & Yáñez Mó, 2014*).

Exosome secretion from the cell is mediated by trafficking proteins. Rab GTPase is involved in intracellular vesicle translocation and trafficking MVB to the plasma membrane for exosome release (*Hsu et al., 2010*; *Ostrowski et al., 2010*). Inhibition of Rab35 results in the intracellular accumulation of vesicles and reduced exosome secretion (*Hsu et al., 2010*). The soluble N-ethylmaleimide–sensitive factor attachment protein receptors (SNAREs) complex is required for MVB fusion with the plasma membrane (*Zhao, Holmgren & Hinas, 2017*). Wnt-containing exosomes cannot be secreted without the YKT6 SNARE (*Gross et al., 2012*). Exosome budding and release may require the actin cytoskeleton and microtubule network (*Mathieu et al., 2019*).

Exosomes act on the surface of receptor cells and deliver molecules that affect receptor cell function. Prostate cancer (PC) cell–derived exosomes carry PD-L1, which binds to PD-1 on the surface of effector T cells and inhibits T cell activation (*Poggio et al., 2019*). Exosomes derived from breast cancer cells efficiently deliver miR-130 to macrophages, resulting in the upregulation of M1-specific markers and cytokines (*Moradi-Chaleshtori et al., 2021*). Exosomal binding to the surface of recipient cells activates exosome uptake. Endocytosis, the most frequently reported mechanism of exosome uptake, is mediated by clathrin-dependent pathways (for exosomes derived from the endothelial cell *Banizs et al., 2018*) and clathrin-independent pathways (*Mulcahy, Pink & Carter, 2014*). Clathrin-independent pathways include caveolin-mediated uptake (pheochromocytoma PC12 cell–derived exosomal miRNAs) (*Tian et al., 2014*), macropinocytosis (oligodendrocytes-derived exosomes) (*Fitzner et al., 2011*), phagocytosis (K562/MT4 cell–derived exosomes) (*Feng et al., 2010*), and lipid raft–mediated internalization (glioblastoma-derived exosomes) (*Svensson et al., 2013*). Exosomes have excellent cell uptake properties; thus, cell–derived exosomes can be used as vehicles for intervention therapies with ferroptosis inducers/inhibitors, and experimentally engineered exosomes are excellent drug delivery systems for cancer therapy.

The combination of exosomes and ferroptosis opens new strategies for cancer therapy. Current research is focused on three different strategies: (1) promoting exosome-inhibited ferroptosis to enhance the effects of chemotherapy; (2) encapsulating exosomes with ferroptosis inducers to inhibit cancers; and (3) developing therapies that combine exosomal inhibitors and ferroptosis inducers (Fig. 4).

### Promoting exosome-inhibited ferroptosis to enhance chemotherapy

Exosomes cause chemoresistance in several cancers, and the underlying mechanism involves ferroptosis. For example, acquired chemoresistance in NSCLC and GC is associated with exosome-induced inhibition of ferroptosis (*Song et al., 2021b*; *Zhang et al., 2020*). The first-line treatment for NSCLC is cisplatin, which induces ferroptosis (*Gridelli et al., 2018*; *Guo et al., 2018*), but long-term cisplatin therapy leads to chemoresistance (*MacDonagh et al., 2018*). One mechanism underlying cisplatin chemoresistance is related to ferroptosis (*Song et al., 2021b*). Transfer of cisplatin-resistant NSCLC–derived exosomal miR-4443 to cisplatin-sensitive NSCLC cells upregulates FSP1 expression through METTL3 in an m6A-dependent manner (*Song et al., 2021b*). This process inhibits ferroptosis and

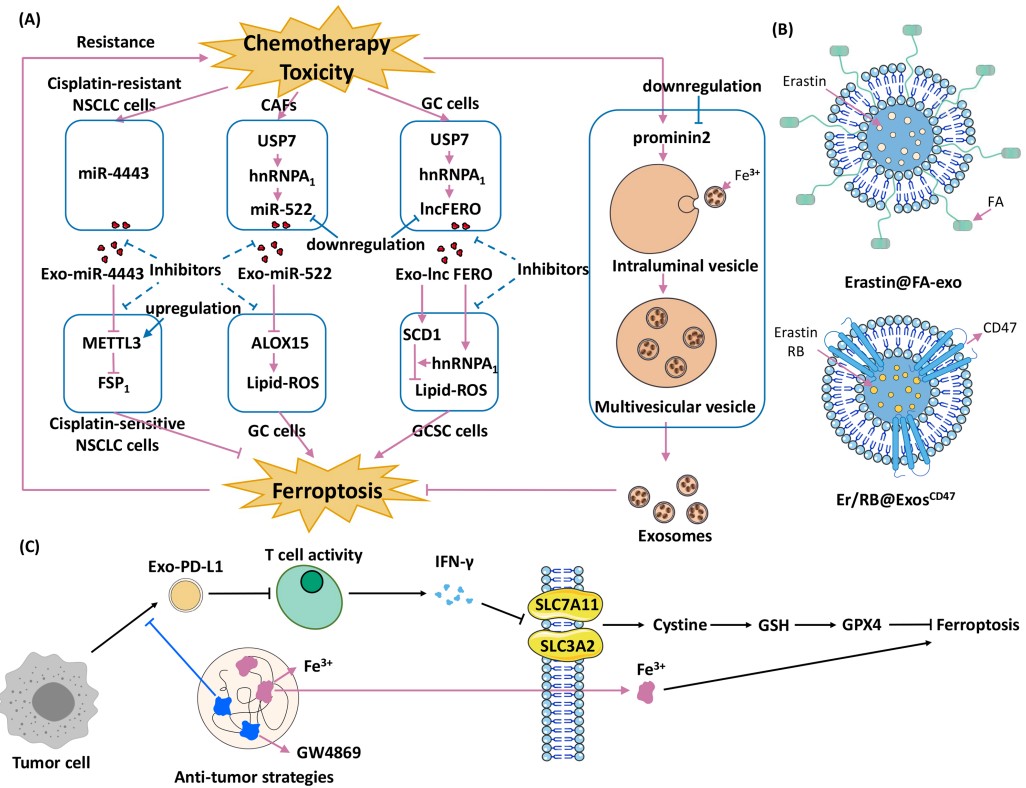

**Figure 4** **Strategies for using exosomes and ferroptosis in tumour therapies.** (A) Promoting exosome-inhibited ferroptosis to enhance chemotherapy. The dotted line indicates the intervention hypothesis, (B) encapsulate exosomes with ferroptosis inducers to inhibit cancers (Erastin@FA-Exo and Erastin /RB@Exos-CD47), (C) develop therapies that combine exosomal inhibitors and ferroptosis inducers (*Wang et al., 2021*).

causes cisplatin-sensitive NSCLC cells to develop resistance to cisplatin (*Song et al., 2021b*). Meanwhile, silencing miR-4443 expression with inhibitors rendered A549-R cells significantly sensitive to cisplatin (*Song et al., 2021b*). Chemotherapy toxicity stimulates the secretion of exosomal miR-522 from CAFs, inhibiting ferroptosis in GC cells and leading to acquired chemoresistance (*Zhang et al., 2020*). Chemotoxicity in CAFs upregulates ubiquitin-specific protease 7 (USP7), which is a drug target for overcoming chemoresistance and antitumour therapy (*Lu et al., 2021*; *Yao et al., 2018*). USP7 regulates deubiquitination of heterogeneous nuclear ribonucleoprotein A1 (hnRNPA1), and elevation of USP7 levels leads to elevated levels of hnRNPA1 (*Zhang et al., 2020*). The hnRNPA1 is involved in the exosomal secretion of multiple miRNAs (*Gao et al., 2019*; *Liu et al., 2021a*). Knockout of USP7 or hnRNPA1 decreased miR-522 levels in the extracellular environment, leading to increased cell death and reduced resistance to chemotherapy (*Zhang et al., 2020*). Another hnRNPA1-associated pathway, involving ferroptosis-associated lncRNA (lncFERO) and SCD1, is also involved in exosome-ferroptosis effects (*Zhang et al., 2021c*). Chemotoxicity targeting ferroptosis promotes exosomal lncFERO (Exo-lncFERO) secretion in GC cells *via* the USP7/hnRNPA1 axis. GC-derived Exo-lncFERO enters gastric cancer stem cells

(GCSCs), binds to SCD1 mRNA and recruits hnRNPA1 to promote SCD1 translation upregulation. This process inhibits ferroptosis and enhances chemoresistance in gastric cancer. Knockdown of hnRNPA1 in GCSCs blocked this effect (*Zhang et al., 2021c*). In summary, cargoes in exosomes, miRNAs and lncRNAs, are responsible for cancer chemoresistance.

Different tumour cell lines have different sensitivities to ferroptosis (*Hangauer et al., 2017*; *Tsoi et al., 2018*). A critical mechanism underlying these differences at the cellular level is caused by differences in prom2 expression (*Brown et al., 2019*). Ferroptotic stress (*e.g.*, interference with GPX4, isolated cells, and extracellular matrix) induces prom2 expression in breast carcinoma cells (*Brown et al., 2019*). prom2Prom2 promotes the binding of intraluminal vesicles to iron-containing ferritin to form ferritin-containing multivesicular bodies. Transporting ferritin out of the cell *via* exosomes inhibits ferroptosis (*Strzyz, 2020*).

Some strategies that promote exosome-inhibited ferroptosis may prevent chemoresistance in cancer cells; for example, (1) inhibiting the secretion of specific exosomes in chemoresistance, (2) regulating FSP1 m6A modification (*Song et al., 2021b*), (3) decreasing the level of miR-522/lncFERO (*Zhang et al., 2020*; *Zhang et al., 2021c*), and (4) targeting prom2 (*Brown et al., 2019*). Promoting exosome-inhibited ferroptosis could be a novel approach to reduce the development of chemoresistance in tumour cells and improve the efficacy of chemotherapy.

### Encapsulating exosomes with ferroptosis inducers to confer anticancer effects in target cells

Experimentally engineered exosomes hold vast therapeutic potential (*Cheng et al., 2018*; *Feng et al., 2021*; *Shi et al., 2020b*). Folic acid (FA)-modified exosomes targeting ferroptosis can be used for clinical applications. FA-modified exosomes containing the ferroptosis inducer erastin (Erastin@FA-Exo) target triple-negative breast cancer cells, and confer antitumour effects (*Yu et al., 2019b*). Erastin induces ferroptosis by inhibiting cystine/glutamate antitransporters (*Kwon et al., 2020*). The Erastin@FA-Exo complex improves the cellular uptake of erastin and suppresses cell proliferation better than Erastin@Exo and free erastin. Erastin@FA-Exo promotes ferroptosis by depleting cellular GSH and overproducing ROS (*Yu et al., 2019b*). Exosomal targeting of ferroptosis combined with immune modification and photodynamic therapy (PDT) effectively induces antitumour effects in HCC cells (*Du et al., 2021a*). PDT is a new tumour treatment that injects a photosensitizer such as rose bengal (RB) to accumulate in tumour tissue. Then, the tumour cells are irradiated with a specific laser wavelength to activate the photosensitizer (*Wang et al., 2014a*), thereby producing monooxygenase ions that specifically destroy tumour cells (*Dolmans, Fukumura & Jain, 2003*). CD47 is loaded into donor-cell exosomes to create Exos-CD47, which effectively evades mononuclear phagocyte–mediated phagocytosis (*Du et al., 2021a*). Erastin and RB were then encapsulated into Exos-CD47 (Erastin/RB@Exos-CD47). Erastin inhibited system Xc- and blocked cystine uptake into the cells, leading to GSH depletion and decreased GPX4 activity (*Cao & Dixon, 2016*). This process significantly increased lipid ROS accumulation, and induced ferroptosis in HCC

cells. Erastin/RB@Exos-CD47 effectively exerts anti-HCC effects in *in vivo* and *in vitro* assays, and has much lower liver toxicity than the control group (Erastin/RB@Exos) (*Du et al., 2021a*).

### Developing therapies that combine exosomal inhibitors and ferroptosis inducers

Complexes containing exosomal PD-L1 derived from tumour cells (*e.g.*, melanoma) suppress T cell activity and lead to resistance to tumour therapy (*Page t al., 2014*; *Poggio et al., 2019*). Ferroptosis is involved in T cell immunity and tumour resistance (*Wang et al., 2019a*; *Wang et al., 2019b*; *Wang et al., 2019c*). Wang and colleagues developed HACA-Fe@GW4869 nanoparticles (HGF NPs), which combine an exosome inhibitor (GW4869) with a ferroptosis promoter ($Fe^{3+}$) to stimulate an antitumour response in melanoma cells (*Wang et al., 2021*). GW4869 inhibits the secretion of exosomal PD-L1, which triggers T cell activation and promotes interferon gamma (IFN-$\gamma$) secretion. Subsequently, SLC7A11 and SLC3A2 in the tumour cell cytosol are inhibited by IFN-$\gamma$, cystine is reduced, GSH levels are decreased, and GPX4 is suppressed, thereby promoting ferroptosis. $Fe^{3+}$ directly promotes ferroptosis. The addition of the ferroptosis inhibitor liproxstatin reduced this effect. Subsequently, this team used modified semiconductor polymers, $Fe^{3+}$, and GW4869 to develop novel phototheranostic metal-phenolic networks (PFG MPNs) (*Xie et al., 2022*). PFG MPNs also possess GW4869 (to block exosomal PD-L1) and $Fe^{3+}$ (to promote ferroptosis) (*Xie et al., 2022*). Moreover, PFG MPNs promote dendritic cells maturation upon integrated laser irradiation (*Xie et al., 2022*), enhancing antitumour therapy. PFG MPNs have excellent near-infrared (NIR) type II fluorescence/photoacoustic imaging performance under NIR laser irradiation (*Xie et al., 2022*). Together with photothermal therapy, PFG MPNs may be used for precise malignancy immunotherapy. These results demonstrate that the combination of exosomes and ferroptosis for tumour therapy has excellent potential for clinical applications (Table 2).

## Current and future concerns in exosomal inhibition of ferroptosis
### Mechanism underlying exosomal inhibition of ferroptosis

The upregulation of GPX4 expression and activity has been reported in ferroptosis, but the effects on other Xc-GSH-GPX4 axis genes, such as SLC7A11 and SLC3A2, have been studied less thoroughly. Exosomal miR-4443 modulates FSP1 m6A modification–mediated ferroptosis and facilitates cisplatin resistance in NSCLC. Furthermore, exosomal miR-4443 may regulate ferroptosis-related genes other than FSP1, so the specific pathways of action remain to be established (*Song et al., 2021b*). There are no reports on exosomal regulation of the third ferroptosis defence system, the GCH1-BH4 axis. ALOX15, the AMPK pathway, and PRDX6 are involved in the exosome-ferroptosis effect, but the other ferroptosis pathways remain to be explored. Future research should conduct a deep pathway study to achieve a more comprehensive molecular understanding of exosomal inhibition of ferroptosis.

**Table 2  The current approaches using exosomal inhibitor plus ferroptosis inducer.**

| Cancer type | Exosome inhibitor | Ferroptosis inducer | Mechanism | Effect | Ref |
|---|---|---|---|---|---|
| melanoma | GW4869 | $Fe^{3+}$ | GW4869 inhibits the secretion of exosomal PD-L1, and $Fe^{3+}$ increases lipid ROS levels, synergistically promoting ferroptosis | induces anti-tumour immune responses | *Wang et al. (2021)*<br>*Xie et al. (2022)* |
| non-small-cell lung cancer | – | erastin | inhibits the system Xc- and block cystine uptake into cells, promoting ferroptosis | sensitizes cancer cells to celastrol | *Liu et al. (2021b)* |
| colorectal cancer cells | – | talaroconvolutin A | downregulates the expression of SLC7A11 and upregulates ALOXE3, promoting ferroptosis | suppresses the growth of cancer cells | *Xia et al. (2020)* |
| uterine serous carcinoma | – | sulfasalazine | inhibits the system Xc- and block cystine uptake into cells, promoting ferroptosis | sensitizes cancer cells to chemotherapy drugs | *Sugiyama et al. (2020)* |
| hepatocellular carcinoma | – | sorafenib | inhibits the system Xc- and blocks cystine uptake into cells, promoting ferroptosis | blocks tumour cell proliferation | *Li et al. (2021d)* |
| colorectal cancer | – | RSL3 | suppresses the KIF20A/NUAK1/Nrf2/GPX4 signaling pathway, promoting ferroptosis | enhances the sensitivity to oxaliplatin | *Yang et al. (2021a)* |
| non-small-cell lung cancer | – | Ginkgetin | induces inactivation of Nrf2/HO-1, promoting ferroptosis | enhances the therapeutic effect of cisplatin | *Lou et al. (2021)* |
| pancreatic cancer | – | MMRi62 | induces degradation of FTH1, promoting ferroptosis | suppresses growth and overcoming metastasis | *Li et al. (2022a)* |
| breast cancer | sulfisoxazole | – | targets ETA and inhibits the secretion of exosomal PD-L1 | induces anti-tumour immune responses | *Im et al. (2019)*<br>*Shin et al. (2022)* |

**Notes.**
-: not containing the component.

PD-L1, programmed cell death-Ligand 1; ROS, reactive oxygen species; SLC7A11, solute carrier family 7 member 11; ALOXE3, arachidonate lipoxygenase 3; KIF20A, Kinesin Family Member 20A; NUAK1, NUAK Family Kinase 1; Nrf2, nuclear factor erythroid-2 related factor 2; GPX4, glutathione peroxidase 4; HO-1, heme oxygenase 1; ETA, endothelin receptor A.

### *Exosome-based drug delivery systems*

Cisplatin-resistant NSCLC–derived exosomal miR-4443 promotes cisplatin resistance in NSCLC by regulating FSP1 m6A modification–mediated ferroptosis (*Song et al., 2021b*). This suggests that future studies could reduce cisplatin resistance and develop new anticancer strategies by restoring METTL3/FSP1–mediated ferroptosis in tumour cells.

Cells release iron-containing exosomes by expressing prom2, which transports iron out of the cell and thereby suppresses ferroptosis (*Strzyz, 2020*). Recent studies have attempted to reverse ferroptosis in cancer cells by inhibiting prom2 transcription

(*Brown et al., 2021*). Heat shock factor 1 (HSF1) positively regulates Prom2 transcription. This suggests that Prom2 transcription may be blocked with HSF1 inhibitors, thereby sensitizing chemoresistant cancer cells to drugs that induce ferroptosis. However, the hypothesis still needs to be studied and more prom2 inhibitors must be tested.

Exosomes have become an active topic of current research as a drug delivery system (*Patil, Sawant & Kunda, 2020*). Studies on exosomes loaded with anticancer drugs targeting ferroptosis are limited to erastin acting on the system Xc−. In the future, additional anticancer drugs may be developed to target different ferroptosis pathways. For example, a first-line therapeutic agent for glioblastoma (temozolomide) may induce ferroptosis by targeting DMT1 expression in glioblastoma cells, which partially inhibits cell growth (*Song et al., 2021a*).

### *Exosomes for cancer diagnosis and prognosis*

Exosomes have an important role in liquid biopsies for early detection and prognosis prediction of cancer (*Li et al., 2021b*). Altered ferroptosis markers in exosomes may be useful biomarkers for cancer screening, such as the early detection of HCC (*Sanchez et al., 2021*) and PC (*Yi et al., 2021*). The lipid composition of HCC and PC cell-derived exosomes is altered, and pathway analysis implicates ferroptosis. Lipidomic profiling in plasma exosomes plays a role in the early detection of HCC in patients with cirrhosis (*Sanchez et al., 2021*), and molecular lipids in urinary exosomes can be used as biomarkers for PC (*Skotland et al., 2017*). Ferroptotic pancreatic ductal adenocarcinoma cells (PDACs) with exosomal KRAS$^{G12D}$ may provide information about the prognosis of pancreatic cancer (*Dai et al., 2020*). Oxidative-stressed PDACs produce autophagy-dependent ferroptosis, releasing KRAS$^{G12D}$, which is packaged extracellularly as Exo-KRAS$^{G12D}$ (*Dai et al., 2020*). Exo-KRAS$^{G12D}$ activates signal transducer and activator of transcription 3 (STAT3)-dependent fatty acid oxidation pathways, polarizing tumour-associated macrophages to an M2-like native phenotype and leading to poor prognosis in pancreatic cancer patients (*Dai et al., 2020*). This implies that binding ferroptosis to exosomes holds potential in liquid biopsies of tumours.

## CONCLUSIONS

Exosomes play an essential role in tumour regulation, which involves ferroptosis. Different tissue-derived exosomes inhibit ferroptosis *via* different pathways, which future work must explore. Exosomal inhibition of ferroptosis drives cancer chemoresistance, and new cancer therapeutic agents combining ferroptosis and exosomes have been reported. Future work on exosomes will open new approaches for developing innovative cancer therapies and leveraging exosome-ferroptosis effects.

### Funding

This work was supported by the Key Research and Development projects in the Sichuan Province (No. 2020YFS0172), the Strategic Cooperation Special Project Sichuan University

& Luzhou City (No. 2021CDLZ-8), The National Natural Science Foundation of China Youth Science Foundation Project (No. 81700941). The funders had no role in study design, data collection and analysis, decision to publish, or preparation of the manuscript.

### Grant Disclosures

The following grant information was disclosed by the authors:
The Key Research and Development projects in the Sichuan Province: No. 2020YFS0172.
The Strategic Cooperation Special Project Sichuan University & Luzhou City: No. 2021CDLZ-8.
The National Natural Science Foundation of China Youth Science Foundation Project: No. 81700941.

### Competing Interests

The authors declare there are no competing interests.

### Author Contributions

- Yixin Shi conceived and designed the experiments, performed the experiments, analyzed the data, authored or reviewed drafts of the paper, and approved the final draft.
- Bingrun Qiu performed the experiments, analyzed the data, authored or reviewed drafts of the paper, and approved the final draft.
- Linyang Huang and Yiling Li performed the experiments, analyzed the data, prepared figures and/or tables, and approved the final draft.
- Jie Lin conceived and designed the experiments, analyzed the data, authored or reviewed drafts of the paper, and approved the final draft.
- Yiting Ze analyzed the data, prepared figures and/or tables, and approved the final draft.
- Chenglong Huang analyzed the data, authored or reviewed drafts of the paper, and approved the final draft.
- Yang Yao conceived and designed the experiments, performed the experiments, authored or reviewed drafts of the paper, and approved the final draft.

### Data Availability

This is a Literature Review.

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
