# Peer review of "Exosomes and ferroptosis: roles in tumour regulation and new cancer therapies"

_PeerJ, doi:10.7717/peerj.13238_

## Round 0.1 · original submission · Major Revisions

All reviewers recommended publication of this manuscript after some revision. I therefore suggest the authors carefully address all reviewers comments before submitting a revised version. A common concern, between 2 out of 3 reviewers, appears to be the lack of consistency in the writing style and the organisation of the review. These two points should be substantially addressed to provide a better description of the reviewed literature on the topic.

Reviewer 1 ·

Basic reporting

In this study, the authors aim to review the exosome-mediated regulation of ferroptosis and its related pathways. Generally speaking, this review is good. However, some improvement will be recommended before it can be accepted for publication. Following are the comments:
1. In this review, language is acceptable.
2. Since this review focuses on the regulation of exosome-mediated regulation on ferroptosis, I would suggest the authors to shorten section one which mainly focuses on the role of exosome in tumors. They don’t need to separate this session into different parts. One comprehensive summary of the current research progresses on how exosome promotes cancer aggressive phenotype both in vitro and in vivo will be fine enough. One summarized chart of how exosomal cargos regulate cancer will be needed. This chart should include type of cargo (lncRNA, miRNA, lipid, protein…), regulated cancer types, regulated biological phenotype (proliferation, migration, invasion, angiogenesis…), related signaling pathway (growth factor pathways…). It would be recommended that the authors provide evidence showing that one specific cargo can regulate aggressive phenotype or tumor formation of at least three types of cancers.
3. The last three section are better-organized. However, some improvements are needed. Some subsections are extremely succinct. For instance, subsection 3.1.2, 3.2.2, 3.3 and 5.3 are too succinct. For the mechanism of inhibiting ferritinophagy, the authors only mentioned the role of NCOA4. I believe, far more regulators are involved in ferritinophagy, besides NCOA4. Subsection 3.1.2, 3.3 and 5.3 have the same issue. More comprehensive reviewing is needed.
4. A chart summarizing the current approaches using exosomal inhibitor plus ferroptosis inducer will be recommended.

Experimental design

Please see details in Basic reporting.

Validity of the findings

Please see details in Basic reporting.

Reviewer 2 ·

Basic reporting

The manuscript submitted by Shi et al. summarizes the role of exosomes during ferroptosis. This is a quite new topic, however, I was only able to find one single publication from one of the two corresponding authors regarding the topic of this review article. I am always a bit sceptic regarding reviews written by non-experts.
The review is interesting and provides novel insights of potential future therapies. There is no problem with language and grammar. Figures are also informative and nice.

Experimental design

The first part of this article summarizes the effects and therapeutic potentials of exosomes on tumors. The second part is dedicted to ferroptosis, especially in tumors. Afterwards, the authors describe how exosomes affect ferroptosis and potential resulting anti cancer therapies.
This is a logic way to organize the review.
References are cited.

Validity of the findings

The aim of the review is cleary reported in the introduction, covered by the text and summarized in the conclusions. In addition, the conclusion names future research areas related to the topic.

Additional comments

Unfortunately, for readers it is not entirely enjoying to read the manuscript. It is quite obvious that different people wrote the text and sometimes it is just a row of sentences that are not clearly connected. It seems that the respective authors summarized one publication within one sentence and the next one within another sentence without connecting these sentences. Therefore, many proteins are not introduced, e.g. ZEB1.
I recommend to improve the text in this regard.

Minor points:
Line 215/216: Loss of xc- leads to ferroptosis because it reduces intracellular levels of cystine and thereby GSH synthesis and GPx4 activity.
Sometimes I got the impression that the authors mixed expression and activity, e.g. why should erastin downregulate GPx4 expression?
The strange acknowledgement should be removed.

·

Basic reporting

The manuscript entitled “Exosomes and ferroptosis: roles in tumor regulation and new cancer therapies” by Yixin Shi et al. comprehensively reviewed the roles of ferroptosis and exosomes in tumor pathogenesis. They summarized the regulations of exosome on ferroptosis, including fenton reaction, ferroptosis defense system, and lipid peroxidation. They also discussed exosome and ferroptosis-based anti-tumor therapies, including improving exosome-inhibited ferroptosis, delivering ferroptosis inducers in encapsulated exosomes, and developing exosomal inhibitors and ferroptosis inducers. This review will be a useful resource for scientists in the research field. The literature references were sufficient and authors provided sound field background. The article is well structured. The figures were presented nicely.

Experimental design

Not applicable.

Validity of the findings

Not applicable.

Additional comments

The manuscript will benefit from moderate language editing, including grammars and scientific writing.

---

## Round 0.2 · accepted · Accept

The Authors have now addressed all reviewers' comments and we feel that the manuscript is now suitable for publication.

Reviewer 2 ·

Basic reporting

The authors addressed all concerns.

Experimental design

-

Validity of the findings

-

Additional comments

-